# Techno-Functional and Sensory Characterization of Commercial Plant Protein Powders

**DOI:** 10.3390/foods12142805

**Published:** 2023-07-24

**Authors:** Kadi Jakobson, Aleksei Kaleda, Karl Adra, Mari-Liis Tammik, Helen Vaikma, Tiina Kriščiunaite, Raivo Vilu

**Affiliations:** 1Center of Food and Fermentation Technologies (TFTAK), Mäealuse 2/4B, 12618 Tallinn, Estonia; kadi.jakobson@tftak.eu (K.J.); karl.adra@tftak.eu (K.A.); mariliis.tammik@tftak.eu (M.-L.T.); helen@tftak.eu (H.V.); tiina@tftak.eu (T.K.); raivo@tftak.eu (R.V.); 2Institute of Chemistry and Biotechnology, Tallinn University of Technology, Akadeemia tee 15, 12618 Tallinn, Estonia; 3School of Business and Governance, Tallinn University of Technology, Akadeemia tee 3, 12612 Tallinn, Estonia

**Keywords:** plant proteins, functional properties, water solubility, water-holding capacity, emulsification, foaming, gelling, visco-thermal analysis, sensory analysis

## Abstract

Many new plant proteins are appearing on the market, but their properties are insufficiently characterized. Hence, we collected 24 commercial proteins from pea, oat, fava bean, chickpea, mung bean, potato, canola, soy, and wheat, including different batches, and assessed their techno-functional and sensory properties. Many powders had yellow, red, and brown color tones, but that of fava bean was the lightest. The native pH ranged from 6.0 to 7.7. The water solubility index was 28% on average, but after heat treatment the solubility typically increased. Soy isolate had by far the best water-holding capacity of 6.3 g (H_2_O) g^−1^, and canola had the highest oil-holding capacity of 2.8 g (oil) g^−1^. The foaming capacity and stability results were highly varied but typical to the raw material. The emulsification properties of all powders were similar. Upon heating, the highest viscosity and storage modulus were found in potato, canola, and mung bean. All powders had raw material flavor, were bitter and astringent, and undissolved particles were perceived in the mouth. Large differences in functionality were found between the batches of one pea powder. In conclusion, we emphasize the need for methodological standardization, but while respecting the conditions found in end applications like meat and dairy analogs.

## 1. Introduction

A growing variety of different plant proteins are used for the development of meat and dairy alternatives, but their properties differ from those of animal proteins and are insufficiently characterized. Furthermore, research and practice show that their properties depend on the plant sources and their variety, growing conditions, processing technology, and purity [1,2,3,4,5]. Extensive mapping of the techno-functional and sensory properties of available plant protein ingredients could certainly support their use.

The important properties for the development of plant-based products include solubility, water- and oil-holding capacities, gelling ability, viscoelasticity, foaming, emulsification, extrudability, cohesion, etc. The nutritional composition and sensory properties like odor, taste, texture, and appearance must be also considered. The relative importance of these properties depends on the product in which they are used. In particular, sausage-type products require ingredients with high gelling and emulsification capacity, while fermented dairy alternatives need high solubility in acidic conditions [6,7,8,9].

Based on protein content, ingredients are roughly classified into flours (<50%), concentrates (50–80%), and isolates (≥80%). Usually, dry fractionation using air or wet extraction with solvents is used to separate proteins from the plant matrices. The dry fractionation technique separates the plant seed components according to their fraction densities. In the wet process, the proteins are first solubilized and then separated using ultrafiltration or precipitation. As a result, protein powders with varying purity and functionality can be obtained [10]. It was shown that using different extraction technologies for pea proteins resulted in protein fractions with varying surface hydrophobicity, gelling capacity, and compressive stress values [11]. In comparison to wet extraction methods, dry fractionation avoids protein denaturation and, thus, tends to preserve the native state and functionality of the proteins [12]. However, denaturation can also be beneficial. Bühler et al. (2020) showed that dry heating of fava bean concentrate reduced its solubility from 53% to 23% but improved its water-holding capacity from 1.3 to 3.1 g g^−1^, which can enhance juiciness in meat substitutes [13]. The knowledge of protein functionality will help the industry to develop processing methods to target improved properties for specific food applications.

In comparison to lab-scale protein isolation, large-scale commercial production of protein ingredients employs harsher but more efficient and economical processing conditions. Unfortunately, this typically leads to greater protein denaturation and changes in the functional properties in comparison to lab-scale production [14,15]. To date, only a limited number of studies have targeted industrial plant protein powders. Ebert et al. (2020) [5] studied several commercial protein concentrates and isolates, but the investigated properties were limited to protein content, protein solubility at native pH, and appearance. The majority of the samples showed solubility below 20%, although pea samples varied from 8% to 50% [5]. Burger et al. (2022) [14] studied the pH-dependent solubility, emulsification, and droplet surface properties of five commercial pea protein powders before and after homogenization. Four samples had very low crude protein solubility of around 10% at pH 7, but homogenization increased it to 20–55% by reducing the surface hydrophobicity. The solubility was pH-dependent and was the lowest at acidic pH levels of 4–5 (around 5–10%) even after homogenization [14]. Zhao et al. (2020) and Ma, Grossmann, et al. (2022) investigated a broader range of techno-functional properties of commercial plant protein powders, including color, solubility, water- and oil-holding capacities, emulsification, foaming, gelling, and pasting, but with a limited number of samples [16,17]. Ma, Grossmann, et al. (2022) found soy protein isolate to have higher water-holding capacity compared to other pulse proteins like fava bean, pea, and lentil. The oil-holding capacity was similar in soy, fava bean, and lentil but lower in pea. The lentil isolate had the lightest color, while pea was the yellowest [17]. Zhao et al. (2020) [16] correlated protein solubility with the emulsification properties and water absorption capacity. In their study, the functionality of soy was close to that of pea, while rice had lower protein solubility, foaming, and emulsification properties [16].

The market for meat, egg, and dairy analogs is rapidly growing, elevating the demand for protein-rich ingredients [18]. Soy and wheat proteins are the most represented plant proteins on the market, due to their low cost and great functionality. However, these proteins have health, environmental, and nutritional concerns [19,20,21,22,23,24]. Consequently, pea protein production has emerged, and a wide variety of pea ingredients are already available on the market [25,26,27,28]. Many other crops (like fava bean, rice, and oat) are also gaining attention due to their sustainability, hypoallergenicity, functionality, etc. [29,30,31,32]. Unlike soy and wheat, which have been in commercial use for decades, these new proteins are less studied, and their functional properties are insufficiently characterized. This hinders the development of new products. Thus, we aimed to comprehensively analyze the sensory and techno-functional properties of a wide selection of commercial plant protein powders, with a focus on potential applications in meat and dairy alternatives.

## 2. Materials and Methods

### 2.1. Commercial Protein Powders

We obtained 24 commercial plant protein concentrates and isolates: 9 pea, 4 oat, 4 fava bean, 2 chickpea, and a selection of individual reference samples of mung bean, potato, soy, wheat, and canola. In addition, the sample set covered different production batches from the same pea and oat products spanning a period of three years. We discuss the samples in more detail in Section 3.1.

### 2.2. Color Analysis

Samples were placed in 8 cm × 8 cm transparent plastic mini-grip bags, forming a 5 mm thick layer. The color was measured at three different spots in the CIELAB color space using an NS810 Spectrophotometer (3nh Shenzhen Threenh Technology Co., Ltd., Shenzhen, China) set to illuminant D65 and 10°.

### 2.3. Water Solubility after Heat Treatment at Native pH and pH 4.5

Water solubility was measured gravimetrically at native pH (without adjustment) and at an acidified pH of 4.5 (common for fermented dairy products) in commercial filtered drinking water, mimicking typical industrial processes. Then, 6% powder dispersions were heat treated at 85 °C for 15 min and cooled down in a water bath to room temperature. After that, the native pH was measured using a SevenGo2 pH meter (Mettler Toledo, Greifensee, Switzerland), and the samples were acidified using 10% lactic acid, emulating dairy fermentation. The dispersions were centrifuged for 15 min at 17,290× *g*, and the supernatants were discarded. The precipitates were washed and centrifuged three times and dried overnight in a thermostat at 105 °C. The acidified samples were washed with acidified water (pH 4.5). Water solubility was calculated from the mass of the dried precipitate relative to the initial powder on a dry weight basis (dwb).

### 2.4. Water Solubility Index, Water-Holding Capacity, and Oil-Holding Capacity

The methods for measuring oil-holding capacity (OHC), water-holding capacity (WHC), and water solubility index at room temperature (WSI) were adapted from Stojceska et al. (2009) [33]. Briefly, 1 g of powder was suspended in 10 mL of distilled water or rapeseed oil, gently mixed for 30 min at room temperature, and centrifuged at 3000× *g* for 15 min; the supernatant was carefully decanted, and the remaining precipitate was weighed. WHC and OHC were expressed as the weight of water or oil held by 1 g of powder (dwb). The water supernatant was collected and oven-dried to calculate the WSI, expressed as the percentage of dissolved solids to the initial powder weight (dwb).

### 2.5. Foam Capacity and Stability

Foams were prepared by dispersing 0.20 g of powder in 20 mL of distilled water, following the protocols of Brishti et al. (2017) and Chandra and Singh (2015), with adaptions [34,35]. The samples were frothed at room temperature inside a 50 mL graduated centrifuge tube using a Polytron PT 2100 homogenizer (Kinematica AG, Malters, Switzerland) equipped with a ⌀ 12 mm probe at a speed of 22,000 rpm for 1 min. Foaming capacity (FC) was calculated by measuring the height of the sample volume before (V_1_) and after (V_2_) frothing and reported as $FC\;\left[\%\right]=\frac{V_{2}-V_{1}}{V_{1}}\times 100$. The change in foam volume (V_t_) after 1 h of standing was recorded. Foam stability was reported as $FS\;\left[\%\right]=\frac{V_{t}}{V_{0}}\times 100$, where V_0_ is the initial volume of the foam.

### 2.6. Emulsion Activity and Stability

Emulsions were prepared by dispersing 0.24 g of powder in 12 mL of distilled water and 12 mL of sunflower oil, following the protocols of Brishti et al. (2017) and Yasumatsu et al. (1972), with adaptations [34,36]. First, samples were mixed at room temperature inside a 50 mL graduated centrifuge tube using a Polytron PT 2100 homogenizer (Kinematica AG, Malters, Switzerland) equipped with a ⌀ 12 mm probe at a speed of 11,000 rpm for 1 min. Subsequently, for the determination of emulsion activity (EA), the samples were centrifuged for 5 min at 1100× *g* at 20 °C. EA was calculated by measuring the height of the emulsified layer (H_1_) and the total height of the liquid (H_T_) and reported as $EA\;\left[\%\right]=\frac{H_{1}}{H_{T}}\times 100$. For the determination of emulsion stability after heat treatment (ES), samples were first heated in a water bath at 80 °C for 30 min, then cooled in an ice-water bath for 15 min, and finally centrifuged for 5 min at 1100× *g* at 20 °C. For the calculation of ES, the height of the emulsified layer (H_2_) was recorded. Emulsion stability was reported as $ES\;\left[\%\right]=\frac{H_{2}}{H_{T}}\times 100$. 

### 2.7. Rheological Properties

Rheological analyses were based on the methods of Onwulata et al. (2014) and Sun and Arntfield (2010), with modifications [37,38].

The sample dispersions (9%) in distilled water were prepared in 50 mL centrifuge tubes and mixed with a Vortex-Genie Mixer (Scientific Industries Inc., Bohemia, NY, USA) for 1 min. The foam was removed, and 18 mL of the sample was poured into a rheometer’s cup.

Gelation behavior was studied with a Physica MCR301 rheometer (Anton Paar, Graz, Austria) equipped with a Peltier C-PTD200 temperature control device and a CC27 concentric-cylinder measuring system. Samples were left to equilibrate at 22 °C for 1 min. Then, the samples were heated over a temperature range of 22–95 °C at a rate of 4 °C min^−1^ while being oscillated at a 1 Hz frequency and 1% strain, which was within the linear viscoelastic region determined in the preliminary tests. A low heating rate is necessary to give protein molecules sufficient time to denature and aggregate [39]. The storage modulus (G′) and loss modulus (G″) were determined as a function of temperature. The samples were run in triplicate, and the maximum value of G′ and the gelling point as a crossover point temperature were reported.

Visco-thermal analysis was performed with a Physica MCR301 rheometer equipped with a C-ETD160/ST electrical temperature control device and a measuring system consisting of a C-CC26/ST aluminum cup and a ST24-2D/2V/2V-30/109 stirrer. To prevent evaporation during the heating cycle, a cover was placed on the top of the cup. The samples were pre-sheared at 25 °C at a rotational speed of 500 min^−1^ for 10 s. Viscosity was measured at a constant rotational speed of 160 min^−1^ throughout five measuring intervals. First, the temperature was kept at 25 °C for 2 min, then it was increased to 85 °C at 12 °C min^−1^, then kept constant at 85 °C for 5 min, followed by cooling to 25 °C at 12 °C min^−1^ and, finally, was kept again at 25 °C for 2 min.

### 2.8. Sensory Evaluation

Sensory analysis was conducted at the Center of Food and Fermentation Technologies (TFTAK, Tallinn, Estonia) in a sensory analysis room. The sensory panel consisted of 9 assessors who had previous sensory training and experience with sensory analysis of various plant protein products. All participants from the sensory panel gave consent to take part in the experiment. The participants were informed in advance of the purpose and the procedures of the study. Taking part in the given study was voluntary, and one could withdraw from the test at any time. The participants were in good health and had no known allergies to the components.

The sensory analysis was conducted according to an internal sensory protocol. The samples were prepared as 6% water dispersions using potable water (Saku Läte OÜ, Saku, Estonia). Then, 60 mL of the sample was dosed into the sniffing glass and covered with a lid to avoid the evaporation of the volatile odor compounds. The water dispersions of the samples were mixed vigorously between the pouring to avoid stratification. An additional plant protein sample (Pea 3) was used as a reference in all sensory sessions. All samples were prepared, served, and evaluated at room temperature.

The order of the samples was randomized according to Williams’ Latin square design and presented in sequentially monadic order to the panelists. Palate cleansing was carried out between the samples using potable water (Saku Läte OÜ, Estonia), unsalted crackers (Pladis Ltd., London, UK), and slices of pears.

In total, three modalities were assessed: odor, taste, and texture. Odor and taste included attributes such as overall intensity, raw material intensity (e.g., cereals, legumes—depending on the main ingredient), and off-odor intensity. Furthermore, taste also included bitterness and astringency. The texture modality included “particle size” and “amount of particles”. Additionally, the panelists had the option to leave comments on each modality in a voluntary text box.

The assessors evaluated odor and taste attributes on a 10-point scale (0–9), where 0 = “none”, 1 = “very weak”, 5 = “moderate”, and 9 = “very strong”. For texture, the scale had different descriptors. For particle size, various points on the scale were described as 0 = “particles missing”, 1 = “very small”, 5 = “moderate”, and 9 = “very big”. The scale for the amount of particles, however, was described as 0 = “particles missing”, 1 = “few particles”, 3 = “some particles”, 5 = “several particles”, 7 = “many particles”, and 9 = “mostly particles”. The sensory evaluation results were collected using RedJade software version 3.0.0 (RedJade Sensory Solutions LLC, Martinez, CA, USA).

### 2.9. Statistical Analysis

All solubility parameters, oil- and water-holding capacities, and gelling analyses were carried out in triplicate. Emulsification, foaming, and visco-thermal analyses were carried out in duplicate. Mean values and standard deviations were calculated. Data visualization and analysis were performed in R 4.3.0 (the R Foundation for Statistical Computing, Vienna, Austria). LOESS (locally estimated scatterplot smoothing) regression was applied to the gelling and visco-thermal results. Spearman’s rank correlations and their significance (α = 0.05) were calculated with the R package “correlation” 0.8.4 and then visualized with the R package “corrplot” 0.92.

## 3. Results and Discussion

### 3.1. Commercial Protein Powders

For our study, we selected products from emerging crops, preferring those that had commercial availability and use. One soy product and one wheat product were selected for comparison. A list of the samples and their protein contents according to the specifications is shown in Table 1. As the functionality of protein powders depends on the crop variety, growing conditions, and processing, we also included different production batches from one pea product and one oat product [4]. In Table 1 and throughout the article, these are marked with asterisks, as well as with green color in figures. The full nutritional and functionality data are available in Appendix A.

According to the technical data sheets, 17 out of 24 products were protein isolates with protein contents ranging from 80% to 90%, while 7 samples were concentrates with protein contents of 56–60%. Most pea samples had lower protein content (80–85%) than the soy isolate. This difference was largely due to the presence of fat in pea isolates (5–9%), as commercially important soybean oil is typically extracted before the soy protein isolation process. The lowest protein content of 56–59% was observed in oat concentrates, which in addition to 9–13% fat contents also had a large share of carbohydrates. Three out of four fava bean samples were also concentrates with protein contents of 60%, which is typical for a dry fractionation process [40]. Overall, the average carbohydrate content was 2% in the isolates and 23% in the concentrates, and the average fat contents were 4% and 7%, respectively. One of the most common parameters provided in the product specification sheets was the pH, which in one case was given as a range of two pH units. Another frequently provided property was the sensory profile, often briefly described as bland, neutral, typical, and characteristic. A few producers stated the country of origin of the plant material, but specific cultivars or varieties were not mentioned. Some specifications included information about the sieve test, color, or descriptive data about the functionality, without including the specific values, methods, and references used in the comparison. Most products, however, had no information about their functional properties, reiterating the need for their characterization.

### 3.2. Color Analysis

The color measurement results in LAB coordinates are illustrated in Figure 1, where bars show the average color of the raw material. A photograph of the powders is shown in Appendix A. Overall, the colors varied considerably between the raw materials, and the pea samples also showed high variance between the products. Lightness values L* (0–100, black–white) were the lowest, at around 74, for potato, canola, and one of the oat samples. The lightest were the fava bean powders, with L* > 90. Lightness was negatively correlated with the green–red a* value (Spearman’s ρ = –0.90), meaning darker powders were more red, with a* values up to 5, but the range of the blue–yellow b* values was higher in comparison (11–26), indicating the presence of yellow pigments in all of the samples. Significant color variation was noticed between the pea batches; in contrast, the oat batches shared the same color, implying a more stable production process. Our results generally corroborate the findings of Ebert et al. (2020), who reported the lowest lightness for canola and similar color coordinates for commercial pea, potato, and wheat protein powders [5]. However, Ma, Grossmann, et al. (2022) reported much lower lightness values of around 50 for commercial pea, soy, lentil, and fava bean powders [17]. Such a large discrepancy can probably be attributed to the differences in the measurement procedures and the equipment used.

Product appearance is the first attribute that consumers assess; therefore, plant protein ingredients should have a light color to facilitate their application in successful product development, especially in dairy replacements [17]. The amount of pigments can vary depending on the plant cultivar and environmental conditions, and they concentrate during the protein isolation process [41,42]. The processing parameters also have a large influence. High temperatures and pH applied during wet production and drying can cause Maillard reactions, caramelization, and oxidation of phenolic compounds and lipids, resulting in a darker color [43]. Washing, antioxidants, defatting, or milder processing conditions during production could be applied to reduce coloration. Furthermore, smaller particle size was shown to increase the lightness of powders [44]. Thus, the high batch-to-batch variation observed with pea products could reflect changes in any of the listed factors.

### 3.3. pH and Solubility

#### 3.3.1. Native pH

The solubility of proteins strongly depends on pH and increases as the pH deviates from the isoelectric point, which for plant proteins is often around pH 4–5 [4]. A notable exception is wheat gluten, which has an isoelectric point around pH 7 [16]. The native pH of our samples in 6% water dispersions after heat treatment was in the range of 6.0–7.7. The highest pH was found in soy. Oat, canola, fava bean, and wheat were mildly acidic at pH 6.0–6.7. Most of the pea samples had pH 7.1–7.6, and large differences between the pea batches were observed, where the pH varied from 6.6 to 7.1. All of the oat batches were consistently at pH 6.6. Ebert et al. (2020) measured the native pH of 3% dispersions without heating and observed even wider pH ranges in commercial protein isolates: pea 5.8–7.7, wheat 5.4–6.1, canola 8.1, and potato 3.3–8.1 [5]. In the dry fractionation process, chemicals are not used, and the pH of the protein concentrate corresponds to the initial pH of the raw material, but in the wet fractionation process based on protein solubilization and precipitation via pH changes, the final pH is chosen by the producer [12]. Because solubility and other functional properties like gelation, emulsification, or extrudability strongly depend on pH, it is important to match the pH of the protein isolate to the final product application, as additional pH adjustments are undesirable [16,45,46]. Furthermore, researchers investigating the functionality of protein isolates in their native state should always report the pH, as it can significantly differ even between batches of the same commercial product. Unfortunately, in many relevant studies, the native pH was not reported [14,16,17,46].

#### 3.3.2. Solubility

We measured powder solubility using three different methods. The water solubility index (WSI) assesses solubility at room temperature and native pH, and this is often used in research, but the other methods apply heat treatment and investigate solubility at two pH points: native and pH 4.5. Moreover, the distinction between whole-powder solubility and protein solubility is important, as protein concentrates (unlike isolates) also contain a significant portion of carbohydrates. In a liquid product, any undissolved particles of the whole powder may sediment, causing sensory defects. Thus, the second and third methods are more relevant to the industry, as products are pasteurized to guarantee microbiological stability, and pH 4.5 is a typical value for fermented dairy analogs.

The WSI results are shown in Figure 2 as dots in various shades of gray and black. Canola and potato protein isolates were almost completely soluble at room temperature, and these products are marketed as such. But in general, the WSI of the powders was below 60%, and on average it was 28%. The lowest WSI (in the range of 4–10%) was observed in wheat, oat, six out of nine pea samples, and one fava bean isolate. The other fava bean samples, which were protein concentrates, had considerably higher WSI of 40–60%. Batch-to-batch WSI variation in pea products was very high (fourfold), but the oat batches were consistent.

The powder solubility after heat treatment is shown in Figure 2 as lines, facilitating the comparison of two pH levels. Solubility at native pH after heat treatment was generally higher than WSI measured at room temperature, but the increase in solubility was greatly dependent on the product. Pea samples were the ones that derived the greatest benefit. The heat treatment step improved the solubility of pea samples at native pH by up to 36%. Pea 5 and soy became 93% soluble, making them especially suitable for liquid applications at neutral pH. Oat, wheat, and chickpea showed only modest improvements of up to 10%, and their solubility remained low. On the other hand, fava bean and mung bean either showed little improvement or even became less soluble. Canola and potato were special cases because heat treatment caused gelation; therefore, their low solubility results instead show that the gels retained the entrapped solutes during the measurement procedure.

As expected, solubility at acidic pH (4.5) after heat treatment was considerably lower in most of the samples. On average, the decrease was threefold, but the sharpest drop was seen in Pea 5 and soy samples, which at their native pH were almost completely soluble. The acidification had little further effect on powders that were poorly soluble at native pH, like oat, mung bean, and some pea and fava bean samples. The only exception was wheat, which became more soluble when the pH shifted away from its isoelectric point at around pH 7.

The differences in powder solubility after heat treatment between the oat batches were insignificant, but the changes between the pea batches were surprising. At native pH, two batches were 13% soluble and two were 37% soluble, but at pH 4.5 the solubility dropped to 7%, except for Pea 8*, which maintained its good solubility and even became the most soluble pea sample in acidic conditions. These large and inconsistent variations between batches pose a challenge for the food industry, as a product developed with one batch might not be possible to replicate with another.

Other researchers have also noted the problematic solubility of commercial plant protein powders, although they have studied crude protein solubility without the heat treatment step. Ebert et al. (2020) found that pea, wheat, rice, sunflower, and pumpkin protein isolates and concentrates had low solubility at pH 7, in the range of 4–50%, unless they were specifically developed to be highly soluble like some potato and canola products [5]. In a study by Burger et al. (2022), five pea samples had only 10% protein solubility at pH 7, but after homogenization this increased to a range of 19–56% [14]. Conversely, at pH 5, the solubility stayed low despite the homogenization process. This confirms that the solubility of plant protein ingredients remains a challenge and requires additional steps like homogenization and heat treatment to improve their functionality. Moreover, some producers of plant protein ingredients recommend these technological steps in their application guidelines.

### 3.4. Water- and Oil-Holding Capacities

Water-holding capacity (WHC) and oil-holding capacity (OHC) describe the ability to hold water or oil during the application of forces [47]. Factors that influence these properties include protein size, amino acid composition, conformation, and carbohydrate and fat contents. Protein powders with more hydrophilic groups have higher WHC due to the formation of a higher number of hydrogen bonds. Likewise, OHC can be explained by hydrophobic and nonpolar side chains on the surface of particles that could interact with oil molecules [48].

We illustrate the WHC and OHC results in Figure 3. Soy had by far the highest WHC of 6.3 g (H_2_O) g^−1^, followed by Pea 5, with a WHC of 3.5 g (H_2_O) g^−1^; both were also notable for their high solubility at native pH. The unmatched WHC of soy in comparison to pea, rice, and wheat was also reported by Zhao et al. (2020) [16]. Pea samples, including different batches of the same product, were highly varied, with WHC values in the range of 0.9–3.5 g (H_2_O) g^−1^. Chickpea, fava bean, mung bean, oat, and wheat tended to have WHC below 2.6 g (H_2_O) g^−1^. Our WHC measurement methodology discards the dissolved part of the sample and considers only the water held by the undissolved part. Consequently, WHC was negatively correlated with WSI among samples with values below 1.3 g (H_2_O) g^−1^ and above 25% WSI. After the exclusion of these samples, the lowest WHC was found in oat and wheat, at 1.5 g (H_2_O) g^−1^. For the same reason, the WHC of canola and potato was not measured, as they dissolved almost completely. In comparison to WHC, OHC was generally lower and with less variation between the samples. The highest OHC was seen in canola and potato, at 2.8 and 2.1 g (oil) g^−1^, respectively. All of the other samples varied between 0.8 and 1.7 g (oil) g^−1^, and the batch-to-batch variation of both pea and oat products was small. Similar to our findings, Fuhrmeister and Meuser (2003) reported the highest WHC in a commercial soy protein isolate, followed by a pea protein isolate, at 4.6 and 4.0 g (H_2_O) g^−1^, respectively [49], but the OHC of their soy (1.2 g (oil) g^−1^) was lower in comparison to our soy sample (1.6 g (oil) g^−1^), and also compared to their commercial pea protein isolate (1.6 g (oil) g^−1^).

The juiciness of plant-based extruded meat analogs was shown to depend on the WHC of the raw material. Furthermore, the generally recommended WHC value for meat analogs is above 3 g (H_2_O) g^−1^ [50]. In our study, only soy and three pea samples were above this threshold, with soy demonstrating a remarkably higher value compared to pea. Our results reveal that in this specific application soy proves to be challenging to replace with alternative protein sources.

### 3.5. Foaming and Emulsification Properties

The foaming and emulsification properties of plant-based ingredients are important for products such as dressings and sauces, as well as alternatives to ice cream, whipped cream, egg, and milk. Adsorption to air–water or oil–water interfaces facilitates the formation and stabilization of gas bubbles or oil droplets by creating a protective coating that generates repulsive forces and introduces some mechanical rigidity that inhibits aggregation [8].

We present the foaming capacity (FC) and foaming stability (FS) results in Figure 4. The FC results were highly variable, covering the whole scale, yet specific raw materials had a characteristic range of values. The highest FC was in wheat (98%) and potato (95%), followed by chickpea and canola, but the lowest was in oat samples (9–19%). The batch-to-batch variation was small. Other studies corroborate the excellent FC of wheat, and that soy has a higher FC compared to pea and fava bean [7,51]. Some reports have attributed high FC to high protein solubility, but our wheat and oat samples both had similarly poor solubility, yet the opposite FC results [52]. Therefore, other factors in addition to solubility are probably more important for foaming, like molecular flexibility and hydrophobicity [53]. The FS results, however, were notably different from the FC results. Foams made with potato and wheat powders were less stable in comparison to soy, which had twofold lower FC. Instead, mung bean, with a low FC of 25%, produced the most stable foam, confirming the results of Tang et al. (2021), who found that mung bean had superior FS compared to soy, chickpea, lentil, pigeon pea, and cowpea [54]. Oat samples, in addition to having the lowest FC, also had FS close to 0%. The FS of pea samples, including different batches, varied in a wide range of 31–68%, but because their FC was relatively low, a higher variance in foam stability measurements was anticipated.

The emulsification activity (EA) and emulsification stability (ES) results are shown in Figure 4. The EA results were mostly around 50%, but oat and mung bean were below 23%, indicating poor emulsification functionality. To confirm these results, we increased the powder concentration during emulsification by 3.5-fold, yet the results were still around 50%. In a study by Tang et al. (2021), a commercial soy protein isolate also had an EA of 54%, much lower than that of their lab-produced isolate with an EA of 71% [54]. Poor emulsification properties of oat and mung bean have also been reported and associated with low solubility [54,55]. Peng et al. (2016) found that pea proteins formed more stable emulsions after heating [56]. In our ES measurement procedure, a heat treatment step was applied, and it generally improved the solubility, but not in mung bean or Oat 1 (Figure 2). We found that heat treatment increased the ES of mung bean to 45% and that of the other oat batches to 35–44% but had no effect on Oat 1. The most notable sample was canola, where the ES improved to 68%. This could be related to its gelling behavior after heating, yet the potato sample that also gelled showed no increase in ES [53]. The ES results of most other samples were in the range of 48–53%, unchanged from the EA. This also confirms that solubility is not the only factor that determines emulsification functionality. The batch-to-batch variation of the EA and ES results between the pea samples was small. Although the variance was higher between the oat batches, this can be disregarded because of their low emulsification functionality.

### 3.6. Visco-Thermal Analysis

A visco-thermal profile reflects the viscosity under a designated heating and cooling regime during constant mixing. This analysis is typically used for assessing the functional properties of starch, which is mainly found in flours and dry-fractionated protein concentrates. In the case of protein isolates, it could show the changes in viscosity due to protein solubilization, denaturation, and gelation after heat treatment and physical processing [37]. In some applications, such as plant-based milk alternatives, the increase in viscosity is considered objectionable. However, in plant-based yogurts and desserts, the viscosity increase is appreciated, and in meat analogs the gelling effect is required [57].

We present the visco-thermal profiles in Figure 5, with the temperature regime outlined by a blue line for guidance. The soy sample had the highest initial (0 min) viscosity of 501 mPa s, followed by a few pea and chickpea samples with viscosities in the range of 119–210 mPa s. For reference, an ordinary dairy yogurt could have a viscosity of around 300 mPa s [58]. The other samples had more modest viscosity values (10–90 mPa s). After heating to 85 °C, at the 12 min point, many samples showed considerable changes in viscosity. The potato sample had the largest viscosity increase, from 13 to 3716 mPa s, which could be related to the gelling behavior that was observed during the solubility analysis. The hot viscosity also increased in mung bean and canola to around 200 mPa s. In contrast, the hot viscosity of some pea samples and soy significantly dropped—in the case of soy, by 24-fold to 21 mPa s. These were the same samples that became considerably more soluble after heat treatment. In the end (19 min), after the cooling step, the viscosity changes in these samples were small. However, mung bean and a few chickpea, fava bean, and pea samples showed an additional increase; thus, their end viscosity greatly exceeded their initial viscosity. The most notable was the potato protein isolate, which after the cooling step achieved the highest overall viscosity of 8857 mPa s. All oat samples, wheat, and some pea, chickpea, and fava bean samples had relatively small viscosity values, making them suitable for applications where high viscosities are not desired. Variation between the oat batches was minimal; in contrast, the pea batches diverged in their behavior during the measurement sequence.

Other studies have used different measurement parameters like powder concentration, temperature, and time, making it difficult to directly compare the results. Osen et al. (2014) measured the visco-thermal profiles of three commercial pea protein powders using different powder concentrations for each of them (15–20%) and a temperature profile of 50–95–50 °C [46]. Two of their pea samples showed a high initial viscosity of around 2000 mPa s, which at 95 °C decreased to 200–350 mPa s, but the third sample had an initial viscosity of around 70 mPa s, which at 95 °C instead increased to 530 mPa s, making the viscosity of all three samples more similar in the end. They related the high initial viscosity to low solubility and high WHC and explained that, upon hydration, the proteins absorb water and swell, but when soluble proteins with low viscosity denature during heating their solubility decreases and, thus, their viscosity increases. Nevertheless, the observations of our results suggested more intricate trends. While the initial viscosity and WHC were indeed strongly correlated (Spearman’s ρ = 0.89), no significant correlation was found between viscosity and WSI. Supporting our results, Webb et al. (2023) demonstrated considerable differences in the visco-thermal profiles between commercial pea products, which could stem from their different cultivars, growing environments, and protein isolation methods [59]. Furthermore, a mixture containing a higher starch content and lower protein content was shown to provide higher end viscosity compared to a mixture containing the highest protein content [60,61]. Conversely, our results showed much lower end viscosity in the starch-containing fava bean concentrates than in the fava bean sample with the highest protein content.

### 3.7. Gelation Behavior

The rheological behavior during heating reveals the thermally induced gelling functionality of powder dispersions in water. Gel formation is important for foods like meat, fish, and egg analogs. Unlike the visco-thermal analysis, during gelation, the sample was maintained in a quiescent state without continuous stirring. The storage modulus G′ represents the solid-like behavior and the loss modulus G″ represents the liquid-like behavior of the sample. When G′ is higher than G″, the sample matrix exhibits gel-like properties, but in this study we consider the structures with G′ < 10 Pa as too weak and, hence, irrelevant in practical applications. For reference, an ordinary dairy yogurt could have G′ on a scale of 100 Pa [62,63].

The majority of our samples had G′ > G″ at 95 °C (Figure 6), indicating more solid-like behavior and the presence of a three-dimensional macromolecular network within the sample. Only the three oat batches and Pea 7* were notable for their more predominant viscous behavior (G′ << G″). The differences between the oat batches were minor, while the pea batches showed more variance. However, overall, almost all of the gels were too weak, as the absolute G′ values were < 10 Pa, except for mung bean, potato, soy, and canola. The G′ value of mung bean gradually increased during heating above 28 °C and reached 1720 Pa, but in potato the increase occurred rapidly between 65 °C and 80 °C, and the highest G′ was 865 Pa—twofold lower than that of mung bean. When these samples were cooled and removed from the rheometer cell, the texture of the gels was firm and, in the case of potato, marmalade-like. Soy performed differently and formed a gel-like texture already at room temperature with G′ = 296 Pa, but during heating the texture became more liquid as both G’ and G″ dropped to 2 Pa. This behavior was similar to the observed viscosity trend in the first half of the visco-thermal analysis. Canola also showed some gelling functionality, but its gel was weak (G′ = 15 Pa), supporting the findings of Yang et al. (2014), who reported similar G′ values at neutral pH and demonstrated that higher pH was needed for stronger thermal gelation of canola [11]. Gelling ability is also stronger at higher protein concentrations in the solution, as it increases the chance of intermolecular interactions, and there is a minimum concentration of protein below which a continuous three-dimensional structure cannot be formed [53]. This could explain the low G′ results of the oat batches, as their protein contents and solubility were also low. The differences in the gelling ability of the same crop could be attributed to the crop variety, protein extraction method, and other components in the powder [64]. Starch and fiber contents can influence the final textural properties of the gel, mainly reducing gel fracture stress and strain, making the gel weaker and more brittle [65]. Hydrocolloids like oat β-glucans, on the other hand, can have synergistic effects with proteins, increasing the gel strength [66,67].

### 3.8. Sensory Evaluation

Unfamiliar and undesirable sensory properties cause lower consumer acceptability of plant-based alternatives [68,69]. Therefore, the development of successful plant-based meat and dairy alternatives requires ingredients with no specific odor or taste. Then, flavor compounds can be added to achieve the desired flavor profile of the final product. However, research shows that plant-based end-products tend to have different off-odors and off-tastes derived from protein powders [70].

As shown in Figure 7, almost all of the evaluated products had an intense overall and raw material odor, with scores in the range of 5.3–8.7. Only chickpea had no specific raw material odor, yet its overall odor was intense. The overall taste intensity and raw material taste attributes were more varied but also quite high (3.4–8.0). In general, chickpea, fava bean, mung bean, and canola had the most intense odor and taste profiles, while soy, oat, wheat, and some pea samples were the least intense. The variation in these taste and odor attributes was evident within the same raw material, e.g., the range of variation between the pea samples was up to 3.2 units. Off-odors and off-tastes were not very apparent (scores < 2.2), since the overall intensity was mostly dominated by the raw material. The exceptions with higher off-flavors were mung bean (rubbery and dusty off-odor, soapy and moldy off-taste) and Pea 9* (sulfurous off-odor).

The samples were rather different in terms of bitterness and astringency, with scores ranging from 1.5 to 5.8, and some characteristics were dependent on the raw material. For example, fava bean samples tended to be generally more bitter and astringent, whereas oat was generally evaluated lower in terms of these attributes. However, bitter compounds vary depending on the plant protein source, and their perception depends on the physiological differences, even among the trained panelists [71]. Thus, consumers’ perceptions of different raw materials may also vary.

The low solubility of plant protein powders was also problematic, because the assessors detected particles in most of the samples (Figure 7). The largest particles, with scores of around 3.0, were found in wheat, mung bean, Pea 1, and Pea 7*. The particle size of the other pea samples varied between 0.9 and 3.0. The samples were more distinguished by the “amount of particles” attribute. The highest scores were given to Oat 1 and mung bean, at 6.7 and 6.3, respectively, which based on the scale translates to between “several” and “many” particles. The “particle size” and the “amount of particles” attributes were correlated (Spearman’s ρ = 0.74); thus, the lowest scores in both attributes were found in soy, canola, potato, and Pea 5 (scores ≤ 1.0). These samples were also the most soluble.

The sensory differences between the oat batches were small, but the pea batches varied greatly, especially in their odor profiles. Two of the pea batches were more intense in odor and also had more off-notes, described as sulfuric and cheesy, while the other two had green, sweet, and rhubarb-like nuances. These characteristics may have influenced the overall sensory profile and, hence, the perception of other attributes like astringency and bitterness, which were also graded higher in these samples. This indicates that the sensory properties of plant protein powders can be influenced by the production method, plant cultivar, and growing conditions [41,70].

### 3.9. Correlations

Figure 8 shows statistically significant Spearman’s rank correlation coefficients, which were multiplied by 100 for brevity. The protein content in the powder had the strongest positive correlations with the WHC (ρ = 0.78), FC (ρ = 0.78), and OHC (ρ = 0.69), and less with the rheological parameters (ρ = 0.45–0.62). This indicates that these functional properties could be attributed more to proteins than carbohydrates, contradicting studies that showed higher viscosity with lower protein and higher starch contents [60,61]. On the other hand, the protein content was not correlated with the three solubility measures, nor with the sensory attributes. This implies that for liquid end-product applications, assessment of just protein solubility instead of whole-powder solubility is insufficient to predict the functional properties of a protein powder, as other constituents could also play a major role.

The native pH and solubility at native pH after heat treatment were positively correlated (ρ = 0.71). Therefore, in the end-products where the pH is not changed, the selection of a protein powder with a higher pH could be more beneficial, as it will be more soluble. Because the native pH of a protein isolate is chosen by the producer, this enables tailoring of the isolation process to specific applications. However, the native pH was not correlated with the WSI, which is measured at native pH and room temperature. Furthermore, the three different solubility parameters were not strongly intercorrelated (ρ ≤ 0.59). This means that to facilitate powder selection, a suitable solubility measurement method must be chosen according to the product being developed, as high cold solubility at native pH does not guarantee that the powder will remain dispersed after product fermentation and pasteurization.

After the exclusion of samples with low WHC and high WSI, as described in Section 3.4, the WHC was found to be strongly correlated with the protein content (ρ = 0.78), native pH (ρ = 0.83), WSI (ρ = 0.83), and solubility at native pH (ρ = 0.70). These findings are in line with the results of Zhao et al. (2020), who studied commercial pea, soy, rice, and wheat protein isolates and concentrates and found a significant correlation between protein solubility and water absorption capacity (Pearson’s r = 0.99) [16]. Regarding the correlation between WHC and protein content, it could be partially explained by the decrease in fat content, as defatting was shown to increase the WHC of flours [72]. The capacity of proteins to bind and retain moisture depends on the type and number of polar hydrophilic groups that are available to bind water. The pH value influences the conformational state of proteins and protein–protein interactions, which, in turn, could hide or reveal those water-binding groups [53]. Surprisingly, the correlation between WHC and solubility at pH 4.5 was negative (ρ = −0.64), probably because samples with a larger number of available polar groups and good water binding aggregated more tightly under acidic conditions and, thus, were less soluble instead. Furthermore, WHC was correlated with the initial viscosity (ρ = 0.89), corroborating the results of Osen et al. (2014), who reported that protein swelling was responsible for the increase in viscosity [73]. However, in our study, the correlations between the WHC and viscosity after the heating and cooling steps decreased to 0.68–0.61. These values are similar to the correlation between water absorption capacity and peak viscosity during heating (Pearson’s r = 0.7) reported by Webb et al. (2023) for commercial pea, soy, and wheat protein powders [59].

The OHC was positively correlated with the protein content and WHC (ρ = 0.69–0.65). Because proteins have hydrophilic and hydrophobic groups, the increase in protein content could have increased the amounts of both groups and enhanced the water and oil binding simultaneously.

The FC was most strongly correlated with the protein content (ρ = 0.78), but not with the three solubility parameters. Potentially, other powder components like fibers could have interfered with the adsorption of proteins to the air–water interface. The FS was most correlated with the WHC and viscosity of the solution; in other words, the FS was higher at lower water mobility conditions. The foaming functionality depends on pH and is the highest at the isoelectric point, where aggregated proteins form thick interfacial films, but in our work the correlation between the native pH and the foaming properties was not observed [53].

Among the visco-thermal parameters, the strongest correlation was found between the hot and end viscosities (ρ = 0.98); hence, their correlations with the other parameters were also similar. However, the initial viscosity before the heating step was different; it was correlated with the WHC and native pH—parameters related to protein swelling. After the heating and cooling steps, however, these correlations decreased. Instead, the hot and end viscosities were most strongly correlated with the storage modulus at 95 °C (ρ = 0.74–0.75); in contrast, the correlation between the storage modulus at room temperature and initial viscosity was low.

Finally, the solubility parameters were negatively correlated with the sensory characteristics of textural granularity. This confirms that plant protein powders must be soluble to a considerable extent for their successful application in dairy alternative products; otherwise, the undissolved particles would be perceived in the mouth, causing sensory defects.

### 3.10. General Discussion

Our results demonstrate that the properties of commercial plant protein isolates and concentrates vary considerably, even within the crop groups. Hence, for successful end-product development, specific protein powders must be selected according to the required functionality. Unfortunately, manufacturers rarely provide this information, making the selection process more challenging.

Typically, studies of protein functionality are performed with protein isolates produced in laboratory facilities employing milder temperature regimes compared to those used in the production of commercial products. Consequently, the properties of laboratory-produced isolates may differ from their commercial counterparts. The current surveys of commercial protein powders include either a small selection of powders or a limited list of investigated techno-functional properties. In addition, various studies employ different measurement procedures and analysis conditions, making comparison of the results more difficult. This implies that methods should be standardized, and the selection of the analysis conditions must be directed by the end applications and consider possible changes that may occur with the proteins during further processing.

We should highlight the lack of strong correlations between the three different solubility measures that we investigated. Heat treatment mostly increased the solubility but also made the method unsuitable for the cases where it caused gelation. The solubility of the protein ingredients generally decreased at pH 4.5, which is particularly relevant to fermented products, but these changes were highly inconsistent between the protein powders, reiterating the need to select the solubility metric according to the end application. In addition, while studies commonly measure the solubility of protein only, the dispersibility of the whole powder cannot be overlooked, as other components like carbohydrates and undissolved particles influence the techno-functional and sensory properties. This is particularly crucial for less-refined protein concentrates.

The color of protein ingredients remains a significant challenge in many applications. Many samples were dark and had red, yellow, or brown tones, which especially limits their use in dairy alternatives but also creates difficulties in achieving meat-like colors during the development of meat analogs. Even though the measurement of color is a straightforward procedure, comparison with the literature can still present challenges. For example, the lightness values reported by Ma, Grossmann, et al. (2022) for commercial powders were 20 units lower than our darkest samples, which were already visually perceived as very dark [17]. Furthermore, Tang et al. (2021) noted that lab-scale powders could be lighter compared to commercial ones due to a milder drying process (lyophilization vs. spray-drying) [54].

All samples exhibited clear raw material flavors, along with bitter and astringent tastes. However, some protein ingredients were significantly more favorable than others, and soy generally emerged as one of the best performers. Different approaches have been proposed to improve the flavor of plant proteins, including cultivar selection and breeding; pretreatments with germination, heating, enzymes, or fermentation; optimization of extraction methods, end-product processing, and storage; taste-masking techniques, and others [74]. Unfortunately, additional processing steps can increase the cost of already-expensive ingredients, particularly when compared to soy.

While the different batches of the same oat sample were consistent, almost all techno-functional properties of the pea batches varied considerably, including native pH, solubility, viscosity, and sensory parameters. Indeed, this poses a challenge for a food production process that has been optimized for a specific batch of plant protein. Therefore, it is essential to monitor the inter-batch functional performance of the plant protein ingredients. WHC, a rather simple analysis, was strongly correlated with many other functional parameters. Thus, WHC, in addition to the native pH and WSI, which can be measured together with WHC, are the simplest and the most accessible methods for monitoring functional performance. Ideally, these parameters could be provided by the manufacturers on their specification sheets as typical functionality indicators of the plant protein ingredients, giving valuable information to their customers.

For future research, we recommend including a wider range of crops and a larger number of representative samples produced using different technologies. To facilitate the comparison between different studies, the characterization methods and their conditions should be standardized, including factors like powder or protein concentration, pH, temperature, and time. We found that native pH has a significant influence on the functional properties of protein powders; thus, future studies should always report it. Furthermore, in the sensory analysis, we focused on a few generalized attributes, but the actual sensory profile of the products was more diverse; therefore, future works could investigate the sensory characteristics in more detail. In addition, our study demonstrated that the acidification and heating steps have a substantial impact on the properties of protein powders, so these treatments could also be incorporated into the sensory and functionality assessment methods.

To conclude, we performed a comprehensive mapping of the techno-functional and sensory properties of commercial plant protein isolates and concentrates. Soy protein isolate demonstrated the best overall properties in our study. However, when considering specific functionality, alternatives to soy protein isolate were also identified. Canola protein isolate had better oil absorption capacity, potato protein formed strong gels, and some pea and fava bean samples were more soluble under acidic conditions (relevant for fermented dairy analogs). Our results highlighted large variations in the properties between different crops, within the crop groups, and between the batches of the same product. Finally, we should emphasize the need for standardized techno-functional characterization methods to support the development of meat and dairy alternatives and to ensure the uniform quality and stability of these products. This, in turn, would contribute to increased adoption and acceptance of novel sustainable food options.

## Figures and Tables

**Figure 1 foods-12-02805-f001:**
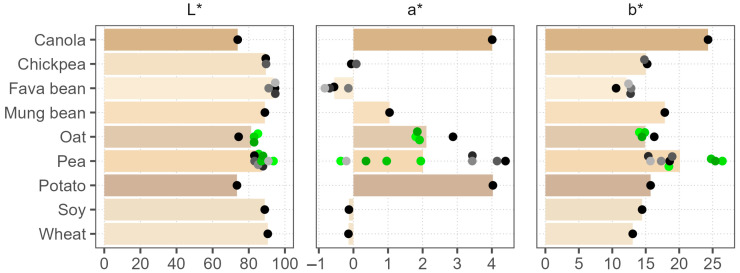
Average powder colors in the CIELAB color space are shown as bar length and bar color. Dots of various shades represent individual products; green color marks different batches of the same product.

**Figure 2 foods-12-02805-f002:**
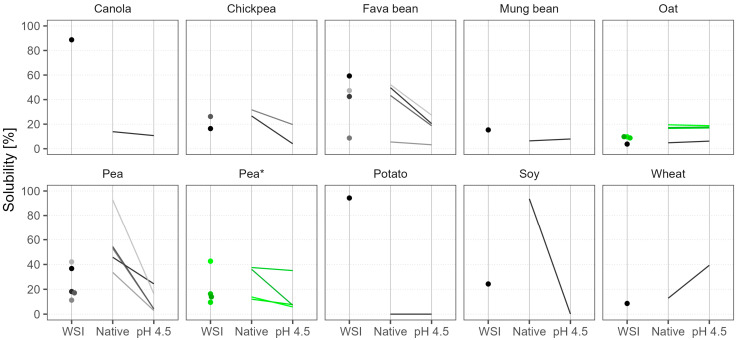
Water solubility of commercial plant protein powders according to three different methods. The water solubility index (WSI) was measured at room temperature and native pH. The other methods involved heat treatment and two pH points: native pH and pH 4.5. Dots and lines of various shades represent different products; green color and the asterisk mark different batches of the same product.

**Figure 3 foods-12-02805-f003:**
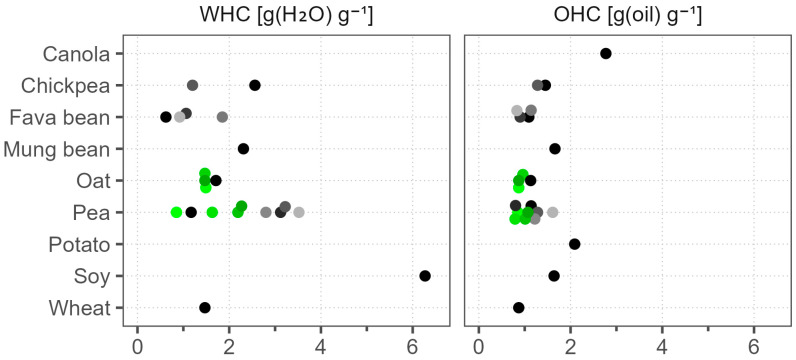
Water-holding capacity (WHC) and oil-holding capacity (OHC) of commercial plant protein powders. Dots of various shades represent different products; green color marks different batches of the same product.

**Figure 4 foods-12-02805-f004:**
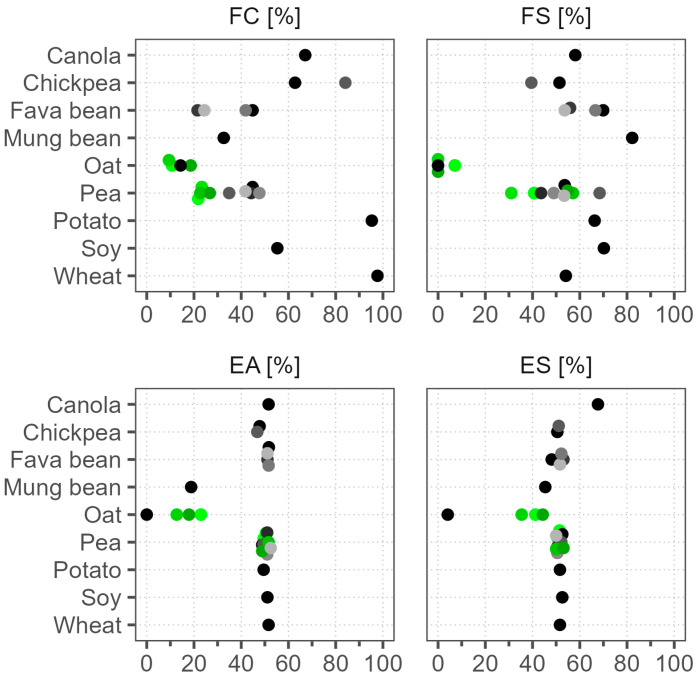
Emulsification and foaming properties of commercial plant protein powders. Dots of various shades represent different products; green color marks different batches of the same product. FC—foaming capacity, FS—foaming stability, EA—emulsification activity, ES—emulsification stability.

**Figure 5 foods-12-02805-f005:**
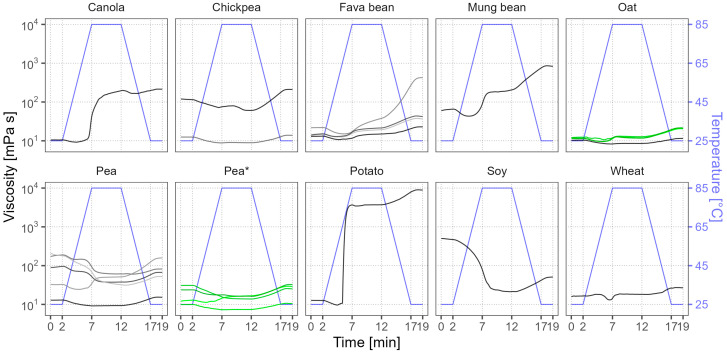
Average visco-thermal profiles of commercial plant protein powders. Lines of various shades represent the viscosity profiles of different samples, and green color and the asterisk mark different batches of the same product. The blue line shows the temperature profile during measurement.

**Figure 6 foods-12-02805-f006:**
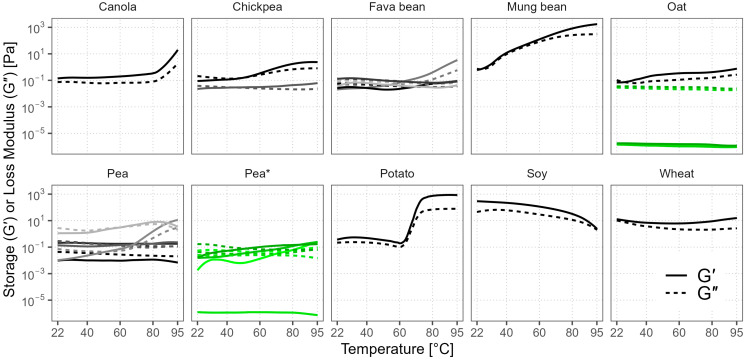
Storage modulus (G′) and loss modulus (G″) of commercial plant protein powders during heating are shown as solid and dashed lines of various shades. Green color and the asterisk mark different batches of the same product.

**Figure 7 foods-12-02805-f007:**
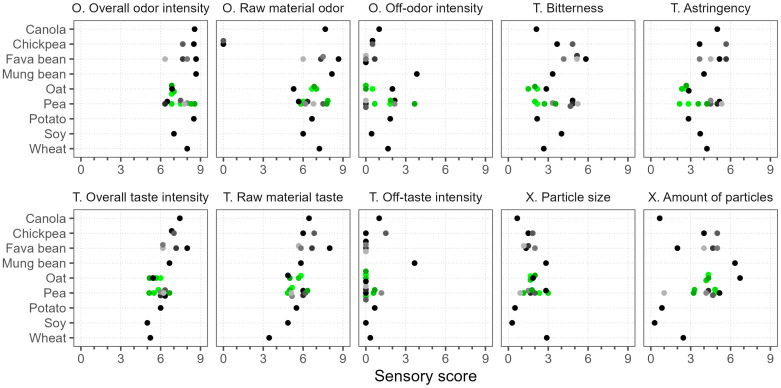
Sensory analysis of commercial plant protein powders. Scale (0–9) is defined as “none”–“very strong” for sensory attributes, “particles missing”–“very big” for particle size, and “particles missing”–“mostly particles” for particle amount. Dots of various shades represent different products; green color marks different batches of the same product. O.—odor, T.—taste, X.—texture.

**Figure 8 foods-12-02805-f008:**
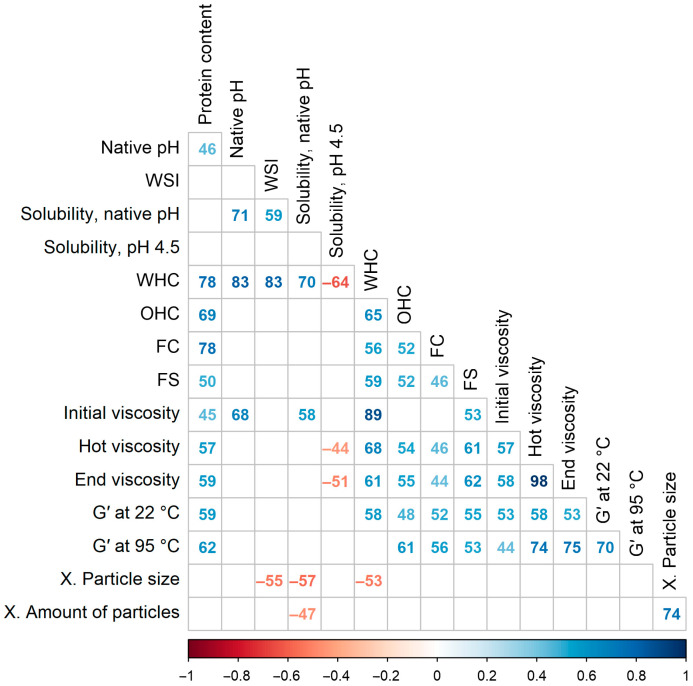
Spearman’s rank correlations between the techno-functional properties of commercial plant protein powders. Multiplied by 100 for brevity. Only statistically significant results are shown (α = 0.05). WSI—water solubility index, WHC—water-holding capacity, OHC—oil-holding capacity, FC—foaming capacity, FS—foaming stability, G′—storage modulus, X.—sensory texture.

**Table 1 foods-12-02805-t001:** Plant protein powder products and their declared protein contents on a dry weight basis (dwb). Samples marked with asterisks are different production batches of the same product.

Sample	Protein Content, % dwb	Sample	Protein Content, % dwb	Sample	Protein Content, % dwb
Canola	90	Mung bean	85	Pea 5	90
Chickpea 1	89	Oat 1	59	Pea 6 *, 7 *, 8 *, 9 *	80
Chickpea 2	89	Oat 2 *, 3 *, 4 *	56	Potato	90
Fava bean 1	60	Pea 1	85	Soy	90
Fava bean 2	60	Pea 2	85	Wheat	82
Fava bean 3	88	Pea 3	84		
Fava bean 4	60	Pea 4	80		

## Data Availability

The data are contained within the article and its Appendix A.

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
