# Peer review of "Techno-Functional and Sensory Characterization of Commercial Plant Protein Powders"

_foods, 2023, doi:10.3390/foods12142805_

Round 1

Reviewer 1 Report

In this manuscript, aothors compared proteins from multiple plants, including pea, oat, fava bean, chick pea, mung bean and so on. Based on the comparison of physical and chemical characteristics between different plant proteins, authors supplies some fundamental data for future utilization into meat and dairy alternatives. This research has applicable significance for food industry. But there are a few questions for authors now.

Firstly, the speices of plant bean is very important. For example, there a lot of species of soy beans around the world. Significant differences exist even among various species of the same bean. So how to confirm if the bean used in this manuscript is typical? Maybe more obvious difference exist between different species of same beans, rather than different beans.

Secondly, meat and dairy alternatives are listed in the topic. But there is no direct experiment that such beans are used in meat and milk preparation. So related experiments should be supplemented, or the topic should be changed.

The manuscript is well written.

Author Response

In this manuscript, aothors compared proteins from multiple plants, including pea, oat, fava bean, chick pea, mung bean and so on. Based on the comparison of physical and chemical characteristics between different plant proteins, authors supplies some fundamental data for future utilization into meat and dairy alternatives. This research has applicable significance for food industry. But there are a few questions for authors now.

Firstly, the speices of plant bean is very important. For example, there a lot of species of soy beans around the world. Significant differences exist even among various species of the same bean. So how to confirm if the bean used in this manuscript is typical? Maybe more obvious difference exist between different species of same beans, rather than different beans.

Answer: we agree, and we mention the importance of plant cultivar in the article. Unfortunately, commercial producers do not state such information. We added discussion on lines 255-256 to reiterate this.

Secondly, meat and dairy alternatives are listed in the topic. But there is no direct experiment that such beans are used in meat and milk preparation. So related experiments should be supplemented, or the topic should be changed.

Answer: as suggested, we shortened the title of the article from “Techno-functional and sensory characterization of commercial plant protein powders for meat and dairy alternatives” to “Techno-functional and sensory characterization of commercial plant protein powders”.

Reviewer 2 Report

The manuscript provides a comprehensive mapping of the techno-functional and sensory properties of commercial plant protein isolates and concentrates. It is an interesting work and can be improved if some corrections were done as follows:

Line 18: if the canola protein was measured, it should be stated in the line 13 too.

Line 81: the authors did not have any comparison with rice bran protein which recently attracted the researches due to hypoallergencity. Please make some comparisons with its functionality. You can find these works: “Rheological properties of dairy desserts: Effect of rice bran protein and fat content” “Physicochemical, functional, and nutritional characteristics of stabilized rice bran form tarom cultivar” “Stabilization of Tarom and Domesiah cultivars rice bran: Physicochemical, functional and nutritional properties”.

Line 82: I know for meat and dairy analog, but no idea for egg. Do you have sample or reference? Please let me know.

Line 90-94: since the researchers used commercial protein powders, it is required to mention the producer and all the details of each protein. Then it can be useful for future studies.

Line 101: Did the authors mean native pH as pH 7.0? why the researchers used pH 4.5?

Line 105: why lactic acid used for decreasing the pH?

Section 2.3: I recommended measuring the protein solubility with the methods of Esmaeili et al., 2016.

Section 2.7: the rheological measurement can be improved through the study “Rheology and microstructure of binary mixed gel of rice bran protein–whey: effect of heating rate and whey addition”.

Line 213 to 220: These statements should be mentioned in the introduction not in the results.

Section 3.2. color: it is useful to provide the photos of all the samples here.

Table 1: I think only protein content is not suffice for the protein specifications. It is necessary to provide the name of supplier, mineral content, moisture and other analysis of the proteins. Then, it can be useful for future works.

Line 233-234: The fat content of the proteins should be analyzed.

Line237: as well carbohydrates?

Line 240-246: the researchers provided the information of the COA (certificate of analysis) which was given by the supplier and each company have different methods in measuring the parameters. It is critical to know where the proteins were provided firstly and then the methods are important. I can not understand how these variations in methods and supplier from different kinds of proteins can be useful?

Line 280: pH measurement, it is essential to use a Table and then have a comparison of the native pH of proteins. Furthermore, the IP of each protein can be useful. However, I have further suggestion for improving the Table, such as molecular weight, intrinsic viscosity and other properties which can be given in this table and used for further studies.

Section 3.4: however the data are presented properly, but for these types of measurements the SD is critical.

Line 449-451: Please provide reference for it.

Line 457: it is redundant to say the viscosity of water and give a reference.

Line 524-531: The work can be cited by “Rheology and microstructure of binary mixed gel of rice bran protein–whey: effect of heating rate and whey addition”

Line 532: For fracture stress and strain, please see this work: Structure-rheology relationships of composite gels: Alginate and Basil seed gum/guar gum.

No specific comments

Author Response

The manuscript provides a comprehensive mapping of the techno-functional and sensory properties of commercial plant protein isolates and concentrates. It is an interesting work and can be improved if some corrections were done as follows:

Line 18: if the canola protein was measured, it should be stated in the line 13 too.

Answer: we added “canola” to Line 14.

Line 81: the authors did not have any comparison with rice bran protein which recently attracted the researches due to hypoallergencity. Please make some comparisons with its functionality. You can find these works: “Rheological properties of dairy desserts: Effect of rice bran protein and fat content” “Physicochemical, functional, and nutritional characteristics of stabilized rice bran form tarom cultivar” “Stabilization of Tarom and Domesiah cultivars rice bran: Physicochemical, functional and nutritional properties”.

Answer: we cited rice protein on Lines 88-89.

Line 82: I know for meat and dairy analog, but no idea for egg. Do you have sample or reference? Please let me know.

Answer: we added a reference to the report by the Good Food Institute https://gfi.org/marketresearch/#eggs

Line 90-94: since the researchers used commercial protein powders, it is required to mention the producer and all the details of each protein. Then it can be useful for future studies.

Answer: additional nutritional information was listed in the supplementary file; we mentioned this on Lines 237-238 and in the back matter section. Because we aimed to characterize the range of current commercial products but not to determine the best and the worst products, we find it unethical to disclose the producers in this study. Nevertheless, our work should stimulate the producers to develop high-quality protein ingredients for special plant-based applications and to provide more functional data on the product sheet.

Line 101: Did the authors mean native pH as pH 7.0? why the researchers used pH 4.5?

Answer: we clarified the terminology and the pH 4.5 choice on Lines 109-110. Additional discussion is present later in the article.

Line 105: why lactic acid used for decreasing the pH?

Answer: we clarified on Lines 114-115 that lactic acid emulates dairy fermentation.

Section 2.3: I recommended measuring the protein solubility with the methods of Esmaeili et al., 2016.

Answer: the suggested method measures only protein solubility, but protein concentrates investigated in our work contain up to 30% of carbohydrates, which also contribute to the overall functionality of a powder. Thus, as we discussed in Section 2.3, we chose methods that are suitable for protein isolates and concentrates both and represent conditions found in food applications.

Section 2.7: the rheological measurement can be improved through the study “Rheology and microstructure of binary mixed gel of rice bran protein–whey: effect of heating rate and whey addition”.

Answer: we added to Lines 164-166 some important points about the rheological measurement procedure from the mentioned article.

Line 213 to 220: These statements should be mentioned in the introduction not in the results.

Answer: these lines were moved to Introduction to Lines 82-91.

Section 3.2. color: it is useful to provide the photos of all the samples here.

Answer: we added a photo to the supplementary file and referenced it in the text on Lines 263-264.

Table 1: I think only protein content is not suffice for the protein specifications. It is necessary to provide the name of supplier, mineral content, moisture and other analysis of the proteins. Then, it can be useful for future works.

Answer: we added the available nutritional information to the supplementary file, but we cannot share the name of the supplier due to ethical issues and commercial interests.

Line 233-234: The fat content of the proteins should be analyzed.

Answer: we put the available information about the fat content to the supplementary file and updated the numbers on Lines 250-252.

Line237: as well carbohydrates?

Answer: we put the available information about the carbohydrate content to the supplementary file and updated the numbers on Lines 250-252.

Line 240-246: the researchers provided the information of the COA (certificate of analysis) which was given by the supplier and each company have different methods in measuring the parameters. It is critical to know where the proteins were provided firstly and then the methods are important. I can not understand how these variations in methods and supplier from different kinds of proteins can be useful?

Answer: we aimed to survey the overall properties of commercial products on the market and investigate what information the suppliers provide. For this purpose, an approximate nutritional composition is sufficient. As we discussed in the article, the properties varied between the powders, yet, some were characteristic for the raw material.

Line 280: pH measurement, it is essential to use a Table and then have a comparison of the native pH of proteins. Furthermore, the IP of each protein can be useful. However, I have further suggestion for improving the Table, such as molecular weight, intrinsic viscosity and other properties which can be given in this table and used for further studies.

Answer: we added the native pH values to Supplementary information. As mentioned in the article, according to other studies the isoelectric point of plant proteins is generally in the range of 4–5, which is confirmed in our work by the decrease in solubility presented in Figure 2.

Section 3.4: however the data are presented properly, but for these types of measurements the SD is critical.

Answer: we added SD values to supplementary information because they were small and would have detracted from the discussion in the main text.

Line 449-451: Please provide reference for it.

Answer: a reference was added.

Line 457: it is redundant to say the viscosity of water and give a reference.

Answer: the mention of water viscosity was removed.

Line 524-531: The work can be cited by “Rheology and microstructure of binary mixed gel of rice bran protein–whey: effect of heating rate and whey addition”

Answer: we understood from the articles cited in this and the next question that hydrocolloids can also have synergistic effects and increase gelling ability, thus, we referred to this fact on Lines 547-549.

Line 532: For fracture stress and strain, please see this work: Structure-rheology relationships of composite gels: Alginate and Basil seed gum/guar gum.

Answer: this was answered together with the previous question.

Round 2

Reviewer 1 Report

The authors have followed all the suggestions, and supplemented some discriptions into this manuscript. Now it can be accepted.

No more comments.